# An Agenda for Research of Uncovered Epidemiological Patterns of Tick-Borne Pathogens Affecting Human Health

**DOI:** 10.3390/ijerph20032206

**Published:** 2023-01-26

**Authors:** Agustín Estrada-Peña, Natalia Fernández-Ruiz

**Affiliations:** 1Department of Animal Pathology, University of Zaragoza, 50013 Zaragoza, Spain; 2Instituto Agroalimentario de Aragón (IA2), 50013 Zaragoza, Spain

**Keywords:** ticks, reservoirs, pathogens, climate gradients, communities of vertebrates, epidemiological relationships

## Abstract

The panorama of ticks and tick-borne pathogens (TBP) is complex due to the many interactions among vertebrates, vectors, and habitats, occurring at different scales. At a broad spatial range, climate and host availability regulate most tick processes, including questing activity, development, and survival. At a local scale, interactions are obscured by a high indeterminacy, making it arduous to record in field surveys. A solid modelling framework could translate the local/regional empirical findings into larger scales, shedding light on the processes governing the circulation of TBP. In this opinion paper, we advocate for a re-formulation of some paradigms in the research of these outstanding cycles of transmission. We propose revisiting concepts that faced criticisms or lacked solid support, together with the development of a conceptual scheme exploring the circulation of TBP under a range of conditions. We encourage (i) an adequate interpretation of the niche concept of both ticks and vertebrate/reservoir hosts interpreting the (a)biotic components that shape the tick’s niche, (ii) an assessment of the role played by the communities of wild vertebrates on the circulation of pathogens, and (iii) the development of new approaches, based on state-of-the-art epidemiological concepts, to integrate findings and modelling efforts on TBP over large regions.

## 1. Introduction

Ticks are prominent vectors of pathogens affecting human health; some of them are shared with pets and/or livestock, while some others are maintained in nature in complex epidemiological cycles [1]. These cycles tend to involve uncounted wild animals and result in intricate patterns of relationships between ticks and competent reservoirs, behind the driving forces of climate and the features of the landscape [2]. An enormous amount of information has been gained through numerous field surveys in many countries, supporting a complex picture of epidemiological relationships. However, research on ticks and tick-borne pathogens (TBP) still miss a solid body of science that could integrate the findings obtained at local/regional scales and project them into a larger area. Modelling approaches seem to be necessary for this purpose if we aim to capture the large area patterns.

The importance of wild vertebrates as hosts for the different stages of ticks and/or acting as reservoirs of TBP is well known [3]. However, scale matters in such research, and therefore, local/regional patterns are difficult to translate into the large view. The driving causes observed behind an epidemiological pattern are not simply “translated” to broader regions, since it is known that the relative composition of communities of wild vertebrate impact the circulation of TBP [4], their prevalence in questing ticks and, consequently, the risk to humans. An integrative framework covering *large regions at high resolution*, aimed at capturing, understanding, and preventing the transmission of TBP, is necessary for the development of protection approaches for human health. Obvious logistic issues prevent coordinated and simultaneous surveying efforts in several countries; these coordinated surveys could provide a comparable set of results suitable for building an integrated modelling that include the impact of weather, vegetation, landscape, deforestation, and the spatially and phylogenetically variable importance of the prevailing vertebrates [5,6]. We believe that the reformulation of some principles behind the epidemiology of TBP affecting human health is necessary. Studies in this direction have begun to emerge [7,8,9,10], improving the identification of ticks and TBP with molecular methods and consolidating techniques allowing for intercomparisons of field studies; they remain the building blocks of any integrative approach. However, studies on ticks have seldom incorporated views from fields of ecology, such as the importance of exploited organisms (hosts), modelling methods, or network analyses. Nowadays, information about the geographical distribution of wild hosts for ticks or competent reservoirs for TBP is massively available, as are datasets containing information about the traits of vertebrates that could influence their ability to circulate TBP [11,12]. Weather data are routinely captured by a cloud of earth-orbiting satellites at an unexpected resolution, very adequate for producing large area estimations of the factors affecting tick and vertebrate physiology. High-resolution maps of vegetation and distribution of vertebrates are available for many regions of the world [13].

This opinion paper aims to propose new ways to consider some of the factors affecting the circulation of TBP (schematized in Figure 1), looking for an agenda for research embedded within the notion of “a community ecology of landscapes” [14]. We aim to elaborate on the current perspectives of the components of the epidemiology of TBP, suggesting new approaches, proposing the rethinking of long-standing concepts, and promoting new methodological approaches to these studies. We are not enumerating actual examples that need a different focus. Instead, we aim to build a proposal for research, looking at concepts and not at cases or patterns occurring at regional scales. Our view of TBP and human health aims to be global, prioritizing topics of research, being confident that we are scratching just the tip of the iceberg.

## 2. The Weather and the Ticks

The weather is the main driver of the physiology of ticks, influencing the activity and survival of questing stages or the development and mortality of moulting ones. The weather also regulates other features of the habitat where ticks thrive, such as vegetal coverage or the presence or abundance of hosts, in part. Therefore, weather drives most of the actions impacting the life cycle of ticks (some other actions, such as deforestation or habitat fragmentation, tend to be human driven). Modelling exercises were soon addressed to capture either the seasonal component of the life cycle, or more recently, to address the probability of presence, also called species distribution models (SDMs). A few researchers have explored other solid approaches to the circulation of TBP, such as the calculation of R_0_ for the tick’s life cycle and hence, the circulation of TBP [15,16]. The SDMs may produce results on the probable distribution of a species if based on a correct selection of variables [17,18,19]. Resulting maps are informative for citizens (i.e., indicating hazardous areas), but research on the topic needs a statistically tractable background. We would address how weather could affect tick’s life cycle only after exploring the conclusions of the mathematical definition of the environmental niche. Modelling the potential distribution of TBP from only the presence/absence data of ticks, or from clinical records in humans, may be unreliable.

Every organism tracks a gradient of restricting variables allowing its persistence in the absence of competitors or predators [20]. Some approaches to tracing the tick environmental niche are based on the presumed capacity of some spatially interpolated variables, averaged for several years, that presumably explain the ecology of every species of tick in every corner of their distribution ranges. The prevailing “dogma” in scientific reports is that ticks’ “suitable habitat” can be deduced from a number of explanatory variables (that sometimes lack ecological meaning, such as the reduction to Principal Components). In this view, results could be immediately plotted into a geographic map displaying the *actual* tick distribution (which is actually a probability of its *potential distribution* derived from the matching with climate variables). We show a basic explanation of this concept in Figure 2, using only two hypothetical environmental variables and also including the importance of vertebrate hosts and climate, aiming to simplify the concept to reinforce the view.

Efforts devoted to comparing the basic components of climate that allow the colonization of an area by ticks over large areas are limited [21]. In any case, the field of ticks suffers from the issues already identified regarding the application of SDMs to the representation of their environmental niche [22], namely (i) the “blind” selection of explanatory variables, without an expert assessment of the ecological importance of each variable, (ii) the tendency to map the potential distribution of a tick without considering the availability of vertebrates (resulting in lack of data about TBP circulation, as we will show later), and (iii) the ***unreliable*** projections of these models into future scenarios without considering climatic, statistical, and/or ecological arguments contrary to its use that have already been analysed [23,24,25,26,27,28]. If the models made with the current climate conditions may be unreliable, how to be confident about the future projections? [26,28]. Further on this, species with widely distributed populations (e.g., *Ixodes ricinus* in Europe) have different limiting factors according to the range considered (Figure 3). Thus, the building of one single model intended to capture the potential distribution of the species, may not reproduce the local limiting variables that affect the spread or survival of a species. As far as we know, this has been explored only for the tick *Rhipicephalus decoloratus* in Africa [29], which does not affect humans. Physiological models, as developed for mosquitoes [30], are still unavailable for ticks. These modelling approaches apply the principles of thermodynamics to organisms, deriving models of processes and their physiological consequences [31]. Physiological models are based on mass and energy gains or losses of the studied organisms and are considered a quantitative leap in the modelling of the distribution and seasonality of mosquito-borne pathogens. Similar models should be addressed for ticks, being probably more complex because of the existence of several stages of the tick’s life cycle that probably react differently to stressors.

We can foresee several topics requiring additional research regarding the climate and the environmental niche of ticks, which we would like to propose for future research, as follows:(i)The tick’s environmental niche is a hypervolume with a mathematical definition. A hypervolume is the intersection of the gradient of several climate variables that define the climate comfort for the tick. The capture and mathematical definition of such a niche for species affecting human health greatly improves calculations of risk and design of major strategies for protecting human health from TBP. To note, studies purely devoted to the evaluation of R_0_, the distribution of host and reservoirs, should also evaluate such a niche. The translation of these data to a map would result in an informative tool, but not in an instrument of decision. We do think that the interpretation of the findings obtained through the analysis of the limiting variables experienced by the tick is a priority.(ii)There is a tendency to use the climate variables “as available” on the Internet without further verification or transformation. This is not a criticism of existing climate datasets, but a call of attention for epidemiologists working on ticks and TBP. For example, the seasons of the year are commonly calculated following astronomical dates; however, ticks do not adhere to an artificial construct that lacks an ecological background. Wettest, warmest, or driest quarters are calculated in a similar way. For example, studies [35] could not find “evidence supporting the superiority of the [current] calculation approach”. The superior performance of an approach based on a harmonic regression of monthly series of climate data, allowing the calculation of daily weather values has been demonstrated and validated [32] and scripts for calculations are freely available [33]. It is important to validate these methods or to provide solid variables that (a) have ecological meaning for ticks, (b) are not self-correlated, and (c) have been selected after an evaluation of the impact of the weather on the modelled tick.(iii)Species with large distributions may have “regional strains”, adapted to the prevailing climate, an extreme demonstrated so far for a few species [28,36]. The training of one single model, later projected to the full range of the species seems to not be the best modelling approach. On the other hand, an “-omics” approach to these hypothetical “races” or “strains” could probably provide key details regarding the transmission of TBP. Local strains of ticks, adapted to the prevailing climate, could gain contact with key reservoirs, circulating strains of a pathogen. As far as we know, this extreme has never been addressed in depth.(iv)Field surveys [37,38] proved that the trends of climate are pushing some tick species out of their “historical” limits or altering their periods of questing on hosts [39]. Nevertheless, studies of the *combined* impacts of the climate, the probable density and distribution of the hosts (that are affected also by the climate), and the landscape transformations, are available only for a few medically significant tick species. We think this is a promising field of research because of the need for a solid framework over which to build a new epidemiological approach.

## 3. Vertebrates: The Neglected Component of the Tick Niche

The vertebrate hosts are essential for the parasitic way of life, and they constitute the “biotic side” of the tick ecological niche. Solid field studies support the notion that the relative abundance of key vertebrates is a decisive factor affecting the circulation of some TBP (i.e., [39]). A community of vertebrates can be defined as a group of *co-occurring* species that share a similar gradient of *environmental conditions*, therefore overlapping their niches in a variable portion (see Figure 2). Communities of species result from environmental filtering, biotic filtering (competition or predation between pairs of species), and other processes such as dispersal, temporal variability, and ecological drift [40,41,42]. If we consider the large scale, such as a continent, a community shows compositional spatial gradients as a response to the climate, food availability, shelter, landscape, vegetation composition, competitors, and/or predators [41]. Communities have relative proportions of species, resulting in a modulating effect of the tick load and supporting the variability and prevalence of TBP in some cases [43] or the circulation of different strains of the same species of pathogens [44,45], because the communities may lack in a variable proportion the necessary competent reservoirs.

Such communities exhibit a *phylogenetic diversity* derived from the mixture of species; we consider such diversity the driving force behind the selective circulation of TBP [46,47]. Therefore, the *relative* composition of vertebrates in a community may lead to drastic changes of its ability to either support tick feeding or circulate pathogens. We encourage to not correlate the community of vertebrates with its richness of species. Instead, the phylogenetic diversity of the whole community, the degree of habitat sharing with the ticks, and the preferences of parasitism of the ticks towards each species of vertebrate, would provide a more balanced view of the epidemiology of TBP [44]. It has been stated that “biotic interactions and environmental filtering shape tick host communities distinctively between specific regions” and “host community composition is an important factor determining the persistence of tick-borne pathogens” [4,48]. A high phylogenetic diversity of vertebrates in a region could allow different species of TBP to be transmitted by a variable number of tick species [44,48] or by one generalist vector, feeding on a large variety of vertebrates. A low phylogenetic diversity of the community of vertebrates (e.g., the dominance of a few species) would probably drive the dominance of one or a few TBP. Changes in the vertebrate’s phylogenetic diversity may result from natural or human actions on the landscape, such as changes in culture patterns, deforestation, or habitat fragmentation [48]. A synthesis seems to be necessary, but how to translate important findings from local surveys to the continental scale is yet unknown. A change of paradigm aiming to a unifying framework is a necessary move forward.

How does climate influence the patterns of the phylogenetic diversity and relative abundance of hosts for ticks or competent reservoirs for pathogens over large regions? How do generalist ticks adapt to changing combinations of hosts? These questions have been only superficially addressed but are of pivotal importance in the prediction of future scenarios of TBP transmission and we aimed to summarize in Figure 4 and Figure 5. Changes of climate may affect the interactions among ticks and vertebrates, promoting changes in the rates of contact between ticks and reservoirs, with the consequent variations in the transmission rates of TBP [7]. If the prevailing climate promotes a change in the phylogenetic composition of reservoirs, some vertebrates involved in TBP circulation may no longer share the habitat with the tick vector. On the other hand, new reservoirs could be over-represented increasing the transmission rates of a TBP (see the hypothetical examples in Figure 4 and Figure 5). Since vertebrates have different capacities to harbour and transmit the pathogens to feeding ticks, and ticks may show preferences to some vertebrates as hosts, such re-organization of the community could change the epidemiological status of a TBP in the area.

In this context, we consider necessary to introduce the concept of “keystone vertebrates” [49], which would contribute to feeding most of the ticks and/or spreading specific TBP. One of the most important challenges in this approach is to understand how losing these keystone vertebrates may subsequently lead to a loss of interactions between the *pathogen* and the tick *vector*. However, the number of vertebrate species examined is still low, and many species of ticks are probably more generalist than previously suspected [50]. Studies on the topic [44,51,52] concluded that the networks of ticks, vertebrates, and pathogens tend to be redundant: the same tick species interacts with several groups of vertebrates; therefore, the lack of one or few vertebrates would do not drive to the exhaustion of ticks and/or TBP in the area.

Another key challenge in the epidemiology of TBP is to predict how the communities of ticks and vertebrates simultaneously respond to spatiotemporal variation in abiotic conditions [53,54,55,56]. The integration of the *simultaneous* modelling of the niche for both vertebrates and ticks is a challenge that would provide a privileged view of the contact rates between vectors and reservoirs. However, for multi-species data sets, fitting many models is required, which makes interpretation challenging and computationally requesting. Multi-species modelling should ideally address the two most important drawbacks of the results produced by environmental models, namely the incorrectly identified organisms and the under-representation of some species because the low probability to observe them (to note, the model should deal with hundreds of species of vertebrates at the continental scale, obtained from compiled datasets that would inevitably contain errors of identification). Joint species modelling evaluates simultaneously the environmental suitability of several species, accounting for phenomena of competition; they can accommodate for effects derived from host’s phylogenetic composition and effects of climate on joint distributions [54,55,56]. A model that accounts for species-to-species associations can be expected to be superior in predicting community-level features [57,58], and therefore, its effects on the circulation of pathogens, on ticks, and its response to stressing variables could be adequately measured. Hierarchical modelling of communities [59] could be of great help in the task.

We consider this a field to have plenty of opportunities for exploring unexpected sources of variability regulating the epidemiological chains of TBP, generating hypotheses about how environmental filtering of vertebrates influence both ticks and TBP. Some topics that we consider necessary to address in a near future are outlined below:(i)It is necessary to continue generating knowledge on tick-hosts associations in the context of the community [60]. Other than prevalence values of *single* pathogens on *each* vertebrate, the focus should be on how a tick species could be “allocated” among the available vertebrates in the community, along the gradients of climate, landscape, and phylogenetic diversity of communities.(ii)There is a growing need for sharing of available datasets of tick distribution data, not restricted only to the medically prominent species. Predictive models of possible occurrence need to be trained with accurate and reliable records and coordinates, which are mainly available for a few medically important tick species.(iii)Researchers on ticks and TBP imperiously need to know the estimated distribution of vertebrates in large areas. Good knowledge exists for countries such as the United States (through the GAP program: https://www.usgs.gov/programs/gap-analysis-project, last accessed on 1 November 2022), but this is lacking for most other countries. We urge to the preparation of digital atlases of distributions (actual and potential) of tick hosts and TBP reservoirs, together with their contributions to the circulation of TBP. The analysis of the resulting communities and the impact of phylogenetic relationships on the TBP circulation is a foreseeable outcome.(iv)We consider it of importance to analyse how changes in the circulation of TBP could happen under several restrictive variables, including both the direct effects of the climate, and the *indirect* effects derived of the lack of co-occurrence among vertebrates. Carefully designed field surveys in near sites exhibiting large changes of habitat (i.e., culture areas versus forest, etc.) or even an exhaustive bibliographic search could provide data, at least for the most studied species of ticks.(v)Although it may seem far from the focus of this paper, the study of the impact of the abiotic gradients of the landscape (climate, fragmentation, deforestation, competence phenomena) on the composition of the vertebrate community is a fundamental step towards a better understanding of the epidemiology of TBP. These studies are an urgent need for epidemiologists, stakeholders, and decision makers.

## 4. Analysing the Tick–Host–Pathogen Relationships

The most direct way of collecting information about the relationships among ticks, vertebrates, humans, and pathogens is through field surveys or laboratory protocols. These methods aim to establish the real potential of either a tick as a vector or of a vertebrate as a reservoir; they are pivotal for any further analysis. These studies have been complemented by comparisons between the empirically observed prevalence of TBP in questing ticks and species-specific biological traits of vertebrates (e.g., size, diet, dispersal, offspring) or the ticks trying to reach a consensus on vectorial or reservoir abilities of ticks or hosts, respectively [50]. Recently, network constructs have been demonstrated to show many features of the associations of communities [51,52].

We ponder that the scarcity of data for many species of ticks can prevent a consolidated framework that ideally would integrate all these data. We proposed that a network construct could adequately represent the relationships among ticks, vertebrates, and pathogens; the examples studied thus far [45,46,51,52] have shown that tick-derived networks are recursive, resilient, coevolved structures (to note, co-evolution is not the same that co-speciation). Recursivity confers strength to networks of ticks, vertebrates, and pathogens [61]. A network could explain the resilience to disturbance of the tick–vertebrate–pathogen interactions, contributing to understanding the impact on the transmission rates of TBP given a certain disturbance. With all the recognized challenges [62,63], networks could be based on the contact rates of vertebrates and ticks as driven by environmental niche constraints. The conclusions obtained from the networks should be validated against the actual prevalence of various TBP under different field conditions. If tested adequately, these methods could result in a *holistic* view of the biotic relationships and the vectorial abilities of ticks assembled from many local studies (see Figure 6).

Notably, any network is based on the number of co-occurrences between pairs of organisms [64]. Imbalanced information would influence the observed interactions. The relative abundances of competent reservoirs, the phylogenetic diversity of the community, and the contact rates with target tick(s) would depend on an adequate input for any further calculation. All these tools already exist, and the next move forward should explore how both worlds (field ecology and modelling approaches of ticks and hosts) could complement each other, according to some basic requirements:(i)We firmly support a modelling approach of the expected environmental niche of vertebrates and tick(s) to fill gaps in not yet surveyed areas and obtain an indirect estimation of the contact rates using networks. Indexes of interactions obtained from these models could be calculated for ecological regions, allowing comparison among results from different regions, and testing its usefulness.(ii)An ecoregionalization of large areas, such as a continent, seems to be necessary for prospective studies of tick-borne pathogens [65]. Such division of the territory should be based on key habitat features (such as temperature, water deficit, forest density, etc.) known to affect both ticks and vertebrates.(iii)Ecological studies on vertebrates suggest that while keystone species’ effects on an organism’s persistence are intuitive, their manifestation in complex natural communities is context-dependent and difficult to predict [66]. An empirical evaluation of this concept using a network of tick-vertebrate-pathogens in a target area has never been performed.(iv)One important challenge in the epidemiology of TBP is to understand whether the network constructs could explain which attributes of the vertebrates regulate their relative competence as reservoirs in situations simulating real communities. This is another promising field of research, bringing together both field and in silico approaches.

## 5. Conclusions

We firmly believe that studies about ticks and transmitted pathogens need a major refocus. We favour a deep integration of available geo-referenced data with explicit mention of the hosts and the prevalence of pathogens detected together with adequate reporting of the status of the ticks (questing or feeding). Linking the epidemiology of TBP with the ecology of wild animals, the impact of the landscape on tick survival, and the effects of the habitat structure on the relative abundance of competent reservoirs are topics open for future research. Modelling the tick’s probable distribution requires homogenization of methods adhering to reliable rules, including a strict definition of an adequate set of variables retaining ecological meaning. For example, a reduction of explanatory variables via principal components reduces collinearity but loses any biological information carried by the original variables. This method (which is statistically correct) favours the production of maps but lacks any biological message.

We propose to explore new approaches, such as joint modelling of communities of vertebrates and contact rates with a target tick, together with a statistical background aimed at converting the outcome of such modelling into epidemiological information. It is necessary to continue with field sampling, improving the already growing available information on the topic; these studies represent the most valuable source for validating other statistically based developments. We propose alternative methods that could be applied to large ranges, supported by records of the vertebrate presence and further modelling of their distribution, together with evaluations of phylogenetic and functional diversity, linked to a network construct as the best way to obtain meaningful indices of tick–host relationships.

How best to achieve this remains an open question deeply related to the way we consider metacommunity-structuring patterns. However, using already existing modelling frameworks, strengthening capacity-building, and encouraging research teams already involved in building ecological knowledge around ticks is a logical step forward. It is just a change of scale.

## Figures and Tables

**Figure 1 ijerph-20-02206-f001:**
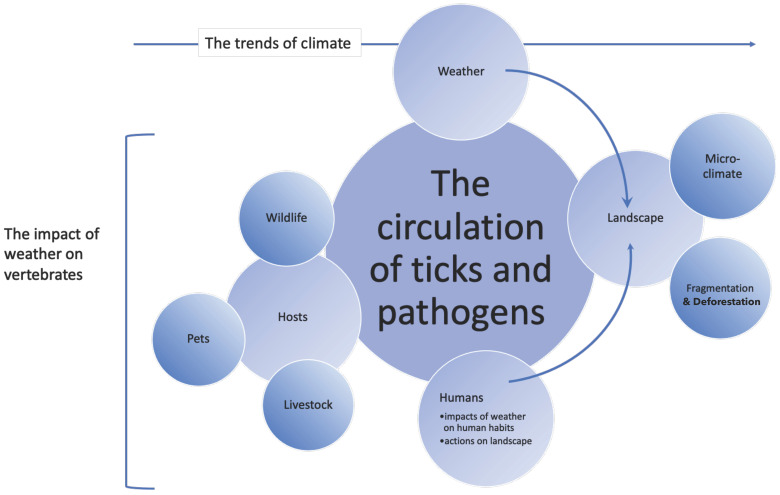
A schematic view of the traits affecting the circulation of tick-borne pathogens. We aimed to represent the main factors that affect the physiology of ticks (survival, development), their involvement with vertebrates carrying pathogens, and their contact with humans while transmitting these pathogens. The “trends of climate” included on top is a generalist term that refers to the many variations (natural or human induced) that are being observed, and that affect the other components. In example, “livestock” may result affected in many ways by the effects of changing climate and, indirectly, have an impact on ticks and transmitted pathogens. In the same way, “humans” may affect the landscape in many ways, which tend to be geographically different (i.e., urbanization). In any case, the chart intends to be only a generalist overview of processes affecting the transmission of pathogens by ticks.

**Figure 2 ijerph-20-02206-f002:**
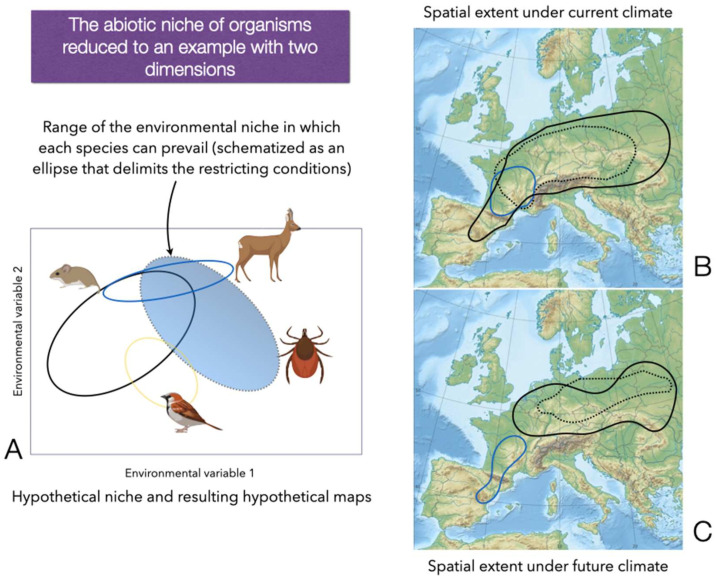
Representation of the concept of “environmental niche”. We illustrate (**A**) a simplified representation of the hypothetical environmental niches of one species of tick and several species of vertebrates, with only two hypothetical variables (axes X and Y). In this view, organisms could have permanent populations in the area inside each ellipse, which “delimits” the area of suitable traits. Areas of overlap among ellipses correspond to areas of “niche overlap” or habitat sharing without geographical barriers. The shared area is thus proportional to the contact rates of a tick with different vertebrates. The suitable niche is commonly translated into maps displaying portions that could fit inside these suitable conditions according to the current climate (contours in (**B**)) or using scenarios of the future climate (hypothetical illustration in (**C**)). Parts of the illustration were created with BioRender.com. Original map of Europe downloaded under licence CC from Wikipedia (https://commons.wikimedia.org/wiki/File:Europe_relief_laea_location_map.jpg accessed on 12 March 2022).

**Figure 3 ijerph-20-02206-f003:**
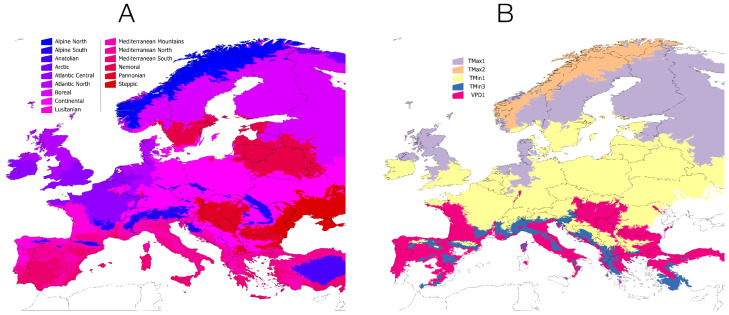
The tick *Ixodes ricinus* has geographically dependent different responses to environmental features. In (**A**), the climate regions of Europe are included with its standard denomination. A total of 11,349 records of the tick was used to develop an environmental suitability model (described in [31,32]). In the case of the illustration in (**B**), we aimed to pinpoint the most limiting variable for tick presence/absence. We did choose as explanatory variables the three first coefficients of a harmonic regression, obtained as explained in [31,32,33]. TMAX, TMIN and VPD refer to maximum temperature, minimum temperature, and vapour pressured deficit, respectively. The numbers next to each variable (1, 2, 3) refer to the annual average (1), the index describing the beginning of the spring (2), or the beginning of autumn (3) from the harmonic regression. See [33,34,35] for complete explanation of the methods that are not the focus of this paper.

**Figure 4 ijerph-20-02206-f004:**
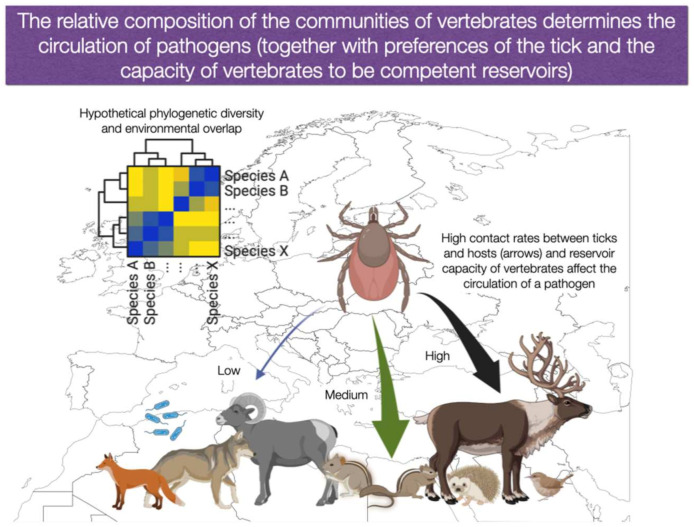
Wild vertebrates belong to communities of different relative composition according to the environmental features, such as climate or vegetation. Ticks have preferences for feeding on different species of vertebrates; in turn, vertebrates have different competence supporting the transmission of pathogens (reservoir capacity). The figure shows a hypothetical community of vertebrates that has a value of phylogenetic diversity and share portions of the habitat; only a few vertebrates hypothetically reservoir a pathogen (left). The transmission of TBP is, in part, due to the traits of each vertebrate and the contact rates of tick vectors with competent reservoirs. In the figure, arrows of different colours and sizes intend to illustrate hypothetical different contact rates among ticks and vertebrates. Thus, the species composition, the relative abundance of each vertebrate, the capacity to reservoir a pathogen, and the pressure of tick bites will shape the permanent circulation of pathogens. Parts of the illustration were created with BioRender.com (accessed on 1 November 2022).

**Figure 5 ijerph-20-02206-f005:**
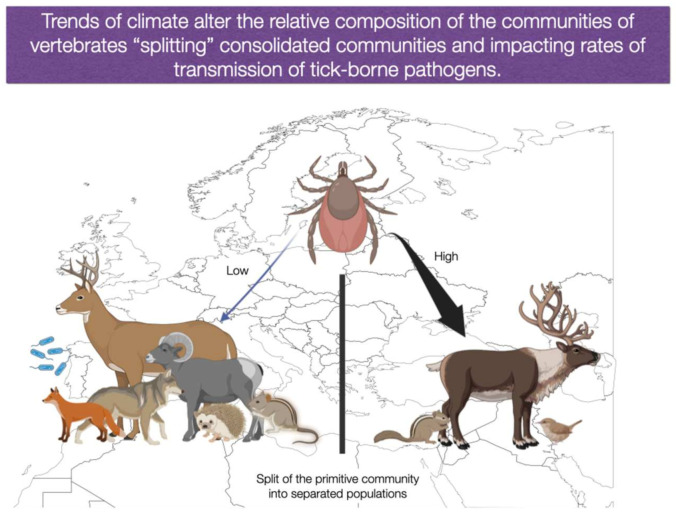
Climate is an essential driver of the reservoir composition of tick-borne pathogens.The situation schematized in Figure 4 is subjected to change because the climate could segregate established communities of vertebrates. Such split of communities could be the outcome of the different tolerance to the climate of the species of vertebrates involved. The split would contribute to changes in the epidemiology of tick-borne pathogens because competent reservoirs could change its density in the newly established communities. The relative abundance of competent reservoirs and contact rates with the tick vector would be different for each group of vertebrates in the new environmental gradient, resulting in new epidemiological patterns. Parts of the illustration were created with BioRender.com (accessed on 1 November 2022).

**Figure 6 ijerph-20-02206-f006:**
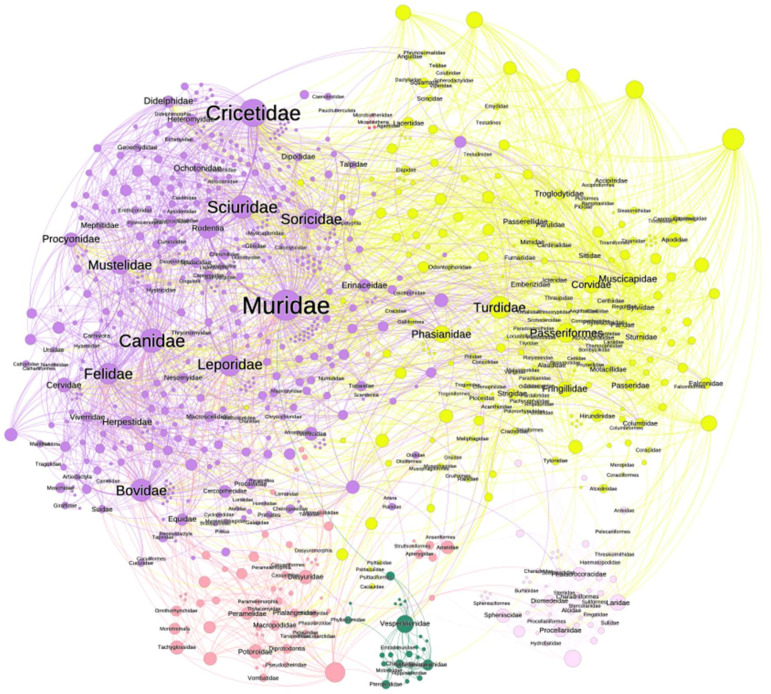
Tools to extract information on the circulation of TBP. Methods used for ecological studies could help to better capture the circulation of TBP. The network displayed shows the reported hosts of every species of the tick genus *Ixodes* in the world (compiled by Alberto A. Guglielmone, INTA, Rafaela, Argentina). Nodes (circles) are either species of ticks or families of vertebrates, and their size is proportional to their importance in the network; the labels of the tick species have been removed to improve the figure’s clarity. Lines (links) represent the use of a host by a tick. The colours correspond to clusters, groups of tick-vertebrates that tend to appear together more frequently than with others; this is a property of the network known as “modularity”. The complete network is available as Appendix A, allowing unlimited on-screen zoom. Such a network could be upgraded with links to pathogens circulated by combinations of vertebrates and ticks and with specific mention to the ecological region. Different indices can be obtained from a network, describing the importance of ecological interactions and co-occurrence events under different scales and abiotic conditions.

## Data Availability

All the material used for this study is available as Appendix A.

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
