# Peer review of "An Agenda for Research of Uncovered Epidemiological Patterns of Tick-Borne Pathogens Affecting Human Health"

_ijerph, 2023, doi:10.3390/ijerph20032206_

Round 1

Reviewer 1 Report

Authors have reviewed and discussed the epidemiological approaches to tick-borne diseases and proposed their opinion. These contents should be necessary and helpful to predict the newly emerging diseases and protect from them. However, it is better to explain with some more concrete examples not just many references.

Author Response

Dear Reviewer, many thanks for your comments. We did our best to provide with more examples, and trying to example some others already existing. The large number of reference is a direct consequence of the many aspects of the issues; we consider that all our comments should be supported by one or several references.

Thank you.

Reviewer 2 Report

Opinion (paper) should be short and clear concentrating on the main subject (modelling). Have used many well-known aspects (fig 1-5, fig 6 difficult readable). Please explain what are unexpected sources of the epidemiological chain of TBPs. Opinion is targeted on the ticks' host vertebrates and very few attention is paid to the influence on human health and public health outcomes. 

Author Response

Dear Reviewer,

Many thanks for your inspiring comments. We aimed to improve some parts of the manuscript that were probably poorly explained. Please note, too, that figure 6 is a large one, and difficult to include in the provided background. Therefore, we opted for including both the "basic figures" and to provide it as Supplementary Material, at high definition, allowing readers an adequate on screen zoom. On the other hand, we modestly consider is focused on the epidemiological aspects of the tick-borne pathogens, not on human health. This focus is mentioned since the beginning of the manuscript.

Thank you.

Reviewer 3 Report

The opinion article is interesting with several important points that are listed by the authors. The scope of the article is the impact of epidemiological aspects on the transmission of diseases by ticks - from animals to humans and the context of ecosystems. All points highlighted by the authors have aspects related to One Health, however these aspects were not well explored and the objective and conclusions are not well structured.

My suggestion in Figure 1 is that, together with "fragmentation", deforestation should be included as one of the main aspects that make up the epidemiological changes that can occur in deforested habitats such as diverse forests spread across continents, such as temperate forests in Europe and tropical forests in Europe, for example. Americas - per example.

Figure 6 is very interesting. However, it is not possible to read some families due to the small font size and how the figure was formatted. Needs to be revised.

Logging and the deforestation of areas in several continents could be more approached by the authors in order to correlate aspects that directly impact the health of ecosystems and the transmission of pathogens.

Once again, we reinforce the need to use One Health concepts that define the interrelationship between environmental health, animal health and human health. This is very important in the spread of parasitic diseases and the tenuous and smooth relationships that can occur in habitats.

Ticks are extremely resistant beings that are possibly capable of surviving in conditions of food scarcity or in regions with severe climatic limitations for survival. The fact is that anthropocentric changes have impacted in an as yet unknown way on the dynamics of pathogen dissemination. For all these reasons I found the opnion article interesting and pertinent with innovative perspectives for the epidemiological studies on the dissemination of arthropods and pathogens that can be transmitted by ticks.

Author Response

Dear Reviewer,

What follows is the verbatim copy of your questions plus our point-by-point replies.

The opinion article is interesting with several important points that are listed by the authors. The scope of the article is the impact of epidemiological aspects on the transmission of diseases by ticks - from animals to humans and the context of ecosystems. All points highlighted by the authors have aspects related to One Health, however these aspects were not well explored and the objective and conclusions are not well structured.

Thank you. We honestly tried to fix the message in our lists(s) of recommendations. Please note, I would not like to call them “conclusions” but just a list of suggestions that follow the comments of the previous section. They are now well structured, some of them are new, and follow a better sort in the text (or we did our best to).

My suggestion in Figure 1 is that, together with "fragmentation", deforestation should be included as one of the main aspects that make up the epidemiological changes that can occur in deforested habitats such as diverse forests spread across continents, such as temperate forests in Europe and tropical forests in Europe, for example. Americas - per example.

Many thanks for this suggestion, that we incorporated into a new figure, repeated the concept in the legend of the figure, and in some other parts of the body of text. We consider it is an important idea, commonly overlooked because it is included in the term “fragmentation” (i.e., a grass replacing an old forest) but the background of the concept is completely different.

Figure 6 is very interesting. However, it is not possible to read some families due to the small font size and how the figure was formatted. Needs to be revised.

Your request It is hard to address because the space left in the template of the journal for figures. We agree with your comments, but please note that, as mentioned, the size of the label has a meaning in the context of the network. Therefore, what we did is to include the same figure as Supplementary Material, at very high resolution, allowing readers to “zoom in” the screen as much as they want. This wa, every label is readable. We tried, too, to make the figure simpler, removing some circles (or nodes). However, this was quite problematic because we were affecting the core structure of the network, and the results were poorer (to state the less).

Logging and the deforestation of areas in several continents could be more approached by the authors in order to correlate aspects that directly impact the health of ecosystems and the transmission of pathogens. Once again, we reinforce the need to use One Health concepts that define the interrelationship between environmental health, animal health and human health. This is very important in the spread of parasitic diseases and the tenuous and smooth relationships that can occur in habitats. Ticks are extremely resistant beings that are possibly capable of surviving in conditions of food scarcity or in regions with severe climatic limitations for survival. The fact is that anthropocentric changes have impacted in an as yet unknown way on the dynamics of pathogen dissemination. For all these reasons I found the opinion article interesting and pertinent with innovative perspectives for the epidemiological studies on the dissemination of arthropods and pathogens that can be transmitted by ticks.

Thank you for your comments. Not only we aimed to re-focus the ideas behind ticks and transmitted pathogens into the context of “One Health” but also to propose some directions in research that we modestly think are of priority. The list, of course, could be longer, but we need to keep the manuscript within some limits…

Reviewer 4 Report

The authors of the paper entitled: "An agenda for research of uncovered epidemiological patterns of tick-borne pathogens affecting human health" proposed an innovative and critical revisitation of key concepts related to the epidemiology of ticks and tick-borne-pathogens affecting human health. The methodological approach explained by the authors is reliable and with robust scientific soundness. 

I have just two minor curiosity/concerns related to this revisiting approach.

1. How much important is the presence of co-exposure of ticks and vertebrates host to different pathogens according to these authors, and how this concern could be introduced and considered in the proposed methodological approach?

2. In my opinion, the introduction of more practical examples and applications of the proposed concepts could capture and hold the reader's attention in a more incisive way.

Author Response

Dear Reviewer,

What follows is your questions and our point-by-point responses. Thank you for your comments and your candid criticism.

  1. How much important is the presence of co-exposure of ticks and vertebrates host to different pathogens according to these authors, and how this concern could be introduced and considered in the proposed methodological approach?
  2. Our opinion is that it is highly important and that it is quite difficult to introduce in the current modelling frameworks, because its complexity. Actually, our recommendations explicitly include a list of actions regarding this topic, that should be urgently addressed.

2. In my opinion, the introduction of more practical examples and applications of the proposed concepts could capture and hold the reader's attention in a more incisive way.

We modestly think that it is difficult. We aimed to increase these practical examples, but they are exactly what we want to demonstrate: that there are highly regional and therefore difficult to integrate into an holistic framework. This is why actual examples "measured" at the regional level are not mentioned (and they are adequate revisions that mention them) because they simply show the contrary we are trying to demonstrate in the Opinion Paper.

Thank you.

Round 2

Reviewer 2 Report

N/A